# The effect of circular stapler size on anastomotic stricture formation in colorectal surgery: A propensity score matched study

**Kil-yong Lee**[1], **Jaeim Lee**[1]*, **Youn Young Park**[2], **Hyung-Jin Kim**[3], **Seong Taek Oh**[1]

**1** Department of Surgery, Uijeongbu St. Mary's Hospital, College of Medicine, The Catholic University of Korea, Seoul, Korea, **2** Department of Surgery, Kyung Hee University Hospital at Gangdong, Kyung Hee University School of Medicine, Seoul, Korea, **3** Department of Surgery, Eunpyeong St. Mary's Hospital, College of Medicine, The Catholic University Hospital, Seoul, Korea

* lji96@catholic.ac.kr

**Data Availability Statement:** The data from this study are available upon request owing to the restrictions (data contain potentially identifying information [birth dates]) imposed by our ethics

## Abstract

### Background

Small circular staplers possess the advantage of being relatively easy to use when compared to larger circular staplers. However, there is some contention as to whether the use of small circular staples in colorectal surgery increases the incidence of anastomotic strictures. This study aimed to determine whether the frequency of anastomosis site stricture formation differs depending on stapler size when performing anastomosis in colorectal surgery.

### Methods

Patients who underwent surgery for colon or rectal disease between June 1, 2009, and December 31, 2021, and who had circular staplers used for the formation of intestinal anastomoses post colectomy were included in our study. Propensity score matching with a 1:1 ratio using logistic regression was performed. The primary outcome was the anastomotic stricture rate, and the secondary outcome was total anastomotic complications.

### Results

A total of 875 patients who were operated on by surgeons using 28/29-mm and 25-mm circular staplers were included. After propensity score matching, 106 patients were assigned to each group. Anastomotic strictures occurred in two cases (1.9%) from the 25-mm group and in four cases (3.8%) from the 28/29-mm group. There were no statistically significant differences between the two groups (p = 0.683). Anastomotic complications were observed in two cases (1.9%) from the 25-mm group and in six cases (5.7%) from the 28/29-mm group; no statistically significant differences were found (p = 0.280).

### Conclusion

Circular stapler size does not influence anastomotic stricture formation in colorectal surgery.

committee (IRB). Please contact our IRB committee for data access via e-mail (irbujb@catholic.ac.kr).

**Funding:** The authors received no specific funding for this work.

**Competing interests:** The authors have declared that no competing interests exist.

## Introduction

When performing colon surgery, the use of a stapler for the creation of intestinal anastomoses has the advantage of allowing easier and faster anastomosis formation than traditional suturing techniques. Moreover, the use of a stapler during anastomosis in right hemicolectomies has been reported to reduce anastomosis site leakage [1]. In colorectal anastomoses, the use of staplers is more convenient than that of traditional techniques due to the narrow pelvic space available [2]. It has been previously established that the rate of anastomotic complications is similar among both techniques; staplers and hand-suturing [3].

Among staplers, the circular stapler can be used when performing end-to-side or end-to-end anastomosis following intestinal resection. It is necessary to advance the circular stapler's body and anvil into the bowel lumen before placing a circular staple. During this process, the width of the intestine, the height of the level of the anastomosis, and the size of the circular stapler are all factors that can result in bowel injury [4, 5]. Therefore, it is necessary to use a circular stapler of an appropriate size. The use of small circular staplers has a disadvantage in that there is a risk of anastomotic site stricture, per previously published literature [6]. However, till date, there are no guidelines for the selection of an appropriate circular stapler size for colorectal surgery. Most of the literature published till date deals only with staplers sized 28/29 mm or more, and there are few reports on the safety of staplers measuring 25 mm [7–11]. Therefore, the purpose of this study was to determine whether the frequency of stricture formation at the anastomotic site varied depending on the size of the staplers used while performing anastomosis in colorectal surgery.

## Methods

This study was approved by the institutional review board (IRB) of the Catholic University of Korea and was performed in accordance with the IRB's guidelines and regulations. All data were fully anonymized before the data assessment. The requirement for informed consent was waived by the IRB to collect data from their medical records used in research.

### Patients

Patients who underwent surgery for colon or rectal disease from June 1, 2009, to December 31, 2021, and had circular staplers used for intestinal anastomosis following colectomy were included. Exclusion criteria were circular stapler sized > 29 mm, lack of colonoscopic follow-up after surgery, and missing data in the covariates used for propensity score matching. The patients were divided into two groups: a group that underwent anastomosis with 25-mm circular staplers and a group that underwent anastomosis with 28/29-mm circular staplers. All data were retrospectively reviewed.

### Procedure

While performing ileocolostomy following ileocecectomy or right hemicolectomy, end-to-side anastomosis was performed using a circular stapler, and a linear stapler was used for colonic stump closure. The anvil of the circular stapler was introduced into the small bowel stump to facilitate anastomosis. During left hemicolectomy, end-to-side or side-to-end anastomosis was performed as part of the colocolostomy. In Hartmann procedure reversal, anterior resection, or low anterior resection, end-to-end anastomosis was performed using the double stapling method. The circular stapler size used was entirely at the discretion of the surgeon.

## Definition

Anastomotic stricture was defined as a case in which a 13.2-mm colonoscope could not pass through or there was resistance to its passage [12]. Therefore, all patients included in this study underwent colonoscopy to check for anastomotic stricture. Anastomotic complications included leakage, bleeding, and stricture formation and were diagnosed based on clinical and radiology findings.

## Outcomes

The primary outcome was the rate of anastomotic stricture formation. The secondary outcomes were rates of anastomotic complications.

## Statistical analysis

For the comparison of clinical characteristics between the two groups (25-mm vs. 28/29-mm), continuous variables were analyzed using the Mann–Whitney U test or Student's t-test, and categorical variables were analyzed using the chi-squared test, Fisher's exact test, or linear-by-linear association.

Propensity score matching with a 1:1 ratio using logistic regression with the nearest-neighbor method was used to correct for differences in baseline characteristics between the two groups. Propensity score matching was conducted using the R package MatchIt (version 3.2.2; R Foundation for Statistical Computing, Vienna, Austria) [13]. The covariates included in the matching were age, sex, height, weight, body mass index (BMI), comorbidities (diabetes, hypertension, heart disease, pulmonary disease, liver disease, and cerebrovascular disease), American Society of Anesthesiologists (ASA) classification, smoking history, alcohol history, emergency surgery requirement, benign or malignant disease, surgical procedure type, and time. After propensity score matching, the clinical characteristics were analyzed using the Mann–Whitney U test or Student's t-test for continuous variables and the chi-squared test, Fisher's exact test, or linear-by-linear association for categorical variables. A series of analyses related to propensity score matching was conducted using Web-based Analysis with R (prepared by web-r.org) based on the R package MatchIt (version 3.2.2; R Foundation for Statistical Computing, Vienna, Austria) [13].

In the analysis of factors affecting the formation of anastomotic strictures, univariate analysis was performed using the chi-squared test or Fisher's exact test by dividing the group into patients with and without anastomotic strictures. For multivariate analysis, only factors with a p-value $<0.1$ upon univariate analysis were analyzed by using a logistic regression test. SPSS version 21.0 for Windows (IBM Corp, Armonk, NY, USA) was used for the comparison of clinical characteristics and the analysis of factors affecting the formation of anastomotic strictures. p-values $<0.05$ were considered statistically significant.

## Results

### Patient selection

From June 1, 2009, to December 31, 2021, 1,689 patients underwent colorectal surgery with anastomosis performed in them using circular staplers. Among the 894 patients who underwent follow-up colonoscopy, 875 patients with 28/29-mm and 25-mm circular staplers were included in the study (Fig 1).

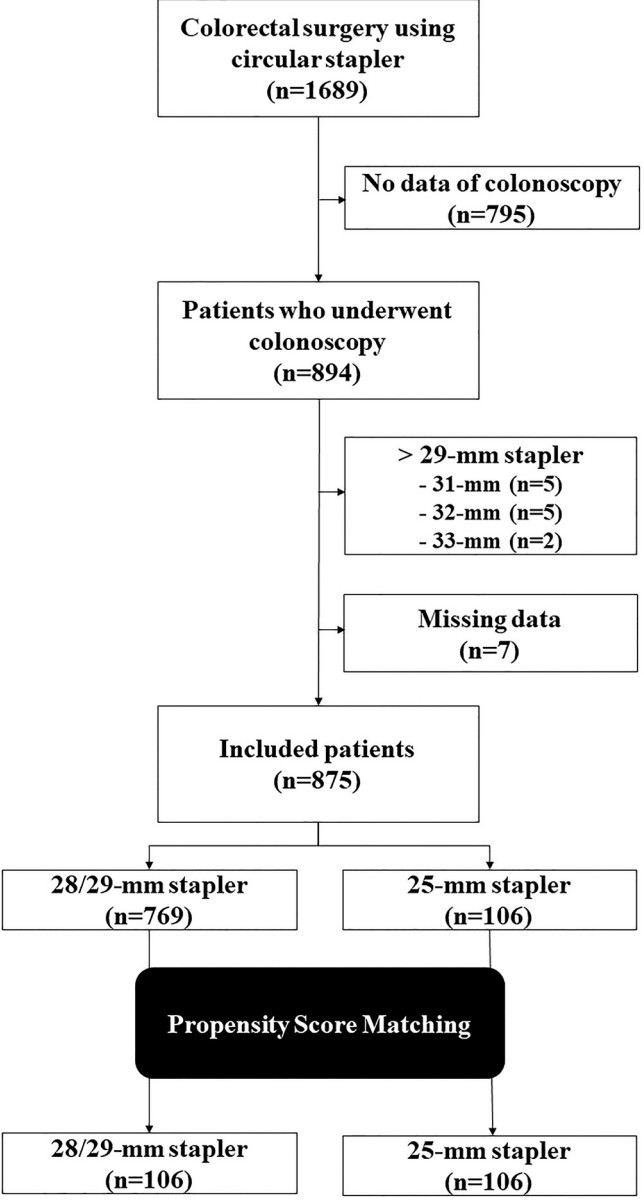

**Fig 1. Flow chart of patient selection.**

## Baseline characteristics

The average age in years, of patients in the 25-mm group and 28/29-mm group was 64.2 and 64.4, respectively (p = 0.888). There were no differences in BMI (p = 0.731) or ASA classification (p = 0.716). However, the 25-mm group had a higher rate of emergency surgery (p<0.001) than the 28/29-mm group, the 25-mm circular staplers were used more frequently in surgeries for benign diseases (p<0.001), and right hemicolectomy or ileocecectomy was more frequently performed in patients (p<0.001). Propensity score matching was performed to correct for these factors, and 106 patients were assigned to each group (Table 1). Following propensity score matching, the factors that showed differences between the two groups were adequately corrected. After surgery, a follow-up colonoscopy was performed at a median of

**Table 1. Clinical characteristics of the two groups before and after propensity score matching.**

| Variables | Before Propensity Score Matching | | | | After Propensity Score Matching | | | |
|---|---|---|---|---|---|---|---|---|
| | 28/29 mm | 25 mm | p-value | Standardized | 28/29 mm | 25 mm | p-value | Standardized |
| Covariates | N = 769 | N = 106 | | Difference | N = 106 | N = 106 | | Difference |
| Age (years) | 64.2 ± 11.8 | 64.4 ± 13.6 | 0.888 | 0.014 | 61.6 ± 13.1 | 64.4 ± 13.6 | 0.134 | 0.203 |
| Sex | | | 0.073 | 0.193 | | | 0.492 | −0.113 |
| Male | 473 (61.5%) | 55 (51.9%) | | | 49 (46.2%) | 55 (51.9%) | | |
| Female | 296 (38.5%) | 51 (48.1%) | | | 57 (53.8%) | 51 (48.1%) | | |
| Height (cm) | 160.9 ± 8.9 | 159.1 ± 10.3 | 0.093 | −0.173 | 158.9 ± 9.4 | 159.1 ± 10.3 | 0.875 | 0.021 |
| Weight (kg) | 62.1 ± 11.4 | 60.1 ± 10.8 | 0.083 | −0.189 | 61.3 ± 10.8 | 60.1 ± 10.8 | 0.399 | −0.116 |
| Body mass index (kg/m$^2$) | 24.0 ± 4.0 | 23.9 ± 4.4 | 0.731 | −0.033 | 24.4 ± 4.6 | 23.9 ± 4.4 | 0.369 | −0.126 |
| Diabetes | 167 (21.7%) | 23 (21.7%) | 1 | 0 | 26 (24.5%) | 23 (21.7%) | 0.745 | −0.069 |
| Hypertension | 352 (45.8%) | 49 (46.2%) | 1 | 0.009 | 49 (46.2%) | 49 (46.2%) | 1 | 0 |
| Heart disease | 47 (6.1%) | 11 (10.4%) | 0.148 | 0.14 | 8 (7.5%) | 11 (10.4%) | 0.631 | 0.093 |
| Pulmonary disease | 40 (5.2%) | 9 (8.5%) | 0.248 | 0.118 | 9 (8.5%) | 9 (8.5%) | 1 | 0 |
| Liver disease | 22 (2.9%) | 3 (2.8%) | 1 | −0.002 | 3 (2.8%) | 3 (2.8%) | 1 | 0 |
| Cerebrovascular disease | 34 (4.4%) | 4 (3.8%) | 0.958 | −0.034 | 8 (7.5%) | 4 (3.8%) | 0.373 | −0.198 |
| ASA classification | | | 0.716 | | | | 0.9 | |
| 1 | 95 (12.4%) | 14 (13.2%) | | 0.053 | 15 (14.2%) | 14 (13.2%) | | −0.049 |
| 2 | 566 (73.6%) | 73 (68.9%) | | −0.008 | 69 (65.1%) | 73 (68.9%) | | 0.061 |
| 3 | 101 (13.1%) | 18 (17.0%) | | 0.126 | 20 (18.9%) | 18 (17.0%) | | 0.022 |
| 4 | 7 (0.9%) | 1 (0.9%) | | 0.287 | 2 (1.9%) | 1 (0.9%) | | 0 |
| Smoking history | | | 0.913 | | | | 0.839 | |
| None | 575 (74.8%) | 79 (74.5%) | | −0.449 | 81 (76.4%) | 79 (74.5%) | | 0.209 |
| Past | 57 (7.4%) | 9 (8.5%) | | −0.444 | 10 (9.4%) | 9 (8.5%) | | −0.104 |
| Present | 137 (17.8%) | 18 (17.0%) | | −0.199 | 15 (14.2%) | 18 (17.0%) | | 0.069 |
| Alcoholic history | 147 (19.1%) | 26 (24.5%) | 0.237 | | 25 (23.6%) | 26 (24.5%) | 1 | |
| Emergency surgery | 37 (4.8%) | 16 (15.1%) | < 0.001 | | 16 (15.1%) | 16 (15.1%) | 1 | |
| Benign or cancerous lesion | | | < 0.001 | | | | 0.189 | |
| Benign | 62 (8.1%) | 30 (28.3%) | | | 40 (37.7%) | 30 (28.3%) | | |
| Cancer | 707 (91.9%) | 76 (71.7%) | | | 66 (62.3%) | 76 (71.7%) | | |
| Operation type | | | < 0.001 | | | | 0.623 | |
| Ileocecectomy or right hemicolectomy | 121 (15.7%) | 55 (51.9%) | | | 46 (43.4%) | 55 (51.9%) | | |
| Left hemicolectomy | 22 (2.9%) | 1 (0.9%) | | | 2 (1.9%) | 1 (0.9%) | | |
| Anterior or low anterior resection | 607 (78.9%) | 47 (44.3%) | | | 55 (51.9%) | 47 (44.3%) | | |
| Subtotal or total colectomy | 19 (2.5%) | 3 (2.8%) | | | 3 (2.8%) | 3 (2.8%) | | |
| Surgery time (min) | 224.4 ± 86.2 | 209.6 ± 74.8 | 0.091 | | 204.4 ± 81.6 | 209.6 ± 74.8 | 0.634 | |

Abbreviation: ASA, American Society of Anesthesiologists

337 days [interquartile range (IQR) 150–753] in the 28/29-mm group and 269 days (IQR 178–386) in the 25-mm group (p = 0.285).

## Outcomes

The primary outcome, anastomotic stricture, occurred in two cases (1.9%) from the 25-mm group and in four patients (3.8%) from the 28/29-mm group. There were no statistically significant differences between the two groups (p = 0.683). It took 131 days (range: 60–271 days) until anastomotic stricture was confirmed.

**Table 2. Anastomotic complications and Clavien–Dindo classification.**

| | 28/29 mm | 25 mm | p-value |
|---|---|---|---|
| Anastomotic Complication | | | |
| Total | 6 (5.7%) | 2 (1.9%) | 0.280 |
| Stricture | 4 (3.8%) | 2 (1.9%) | 0.683 |
| Leakage | 4 (3.8%) | 0 | 0.121 |
| Bleeding | 0 | 0 | > 0.999 |
| Clavien–Dindo Classification | | | 0.429 |
| 0 | 70 (66.0%) | 72 (67.9%) | |
| 1 | 13 (12.3%) | 12 (11.3%) | |
| 2 | 14 (13.2%) | 18 (17.0%) | |
| 3a | 5 (4.7%) | 3 (2.8%) | |
| 3b | 3 (2.8%) | 1 (0.9%) | |
| 4a | 1 (0.9%) | 0 | |

The secondary outcome, anastomotic complications, occurred in two cases (1.9%) from the 25-mm group and in six patients (5.7%) from the 28/29-mm group. No statistically significant differences were found between the two groups (p = 0.280). Anastomotic leakage was observed in four cases (3.8%), all from the 28/29-mm group. None of the patients experienced anastomotic bleeding. There was no difference between the two groups in the Clavien–Dindo classification for overall postoperative complications (p = 0.429) (Table 2).

Upon univariate analysis of factors affecting anastomotic stricture formation, anastomotic leakage (p = 0.004), cerebrovascular disease (p = 0.039), and anastomosis type (p = 0.014) were found to be significant factors. As per the multivariable analysis, anastomotic leakage was found to be the only statistically significant factor [adjusted odds ratio (OR): 39.38, 95% confidence interval (CI): 2.49–623.36, p = 0.009] (Table 3).

## Subgroup analysis

Since all anastomotic strictures occurred after colorectal anastomosis, a subgroup analysis was performed on 105 patients with colorectal anastomosis. No factors in clinical characteristics showed differences between the two groups (S1 Table in S1 File).

Anastomotic stricture occurred in four (7.0%) and two (4.2%) patients in the 28/29-mm and 25-mm groups, respectively, and there was no statistical significance (p = 0.686) (S2 Table in S1 File). There was no difference between the two groups in the anastomotic problem (p = 0.450) and postoperative complications according to Clavien–Dindo classification (p = 0.166).

In the analysis of factors affecting anastomotic stricture, anastomotic leakage was identified as the only influencing factor (adjusted OR: 39.38, 95% CI: 2.49–623.36, p = 0.009) (S3 Table in S1 File).

## Anastomotic stricture management

The three patients in whom anastomotic stenosis was observed, but in whom the colonoscope passed the site of stenosis, albeit with resistance, did not receive any special treatment because there were no overt symptoms of stenosis. However, of the three patients in whom the colonoscope did not pass, interventions were carried out to widen the stricture.

**Table 3. Univariate and multivariate analyses of factors affecting anastomotic strictures.**

| | | Stricture | | Univariate analysis | | | Multivariate analysis | | |
|---|---|---|---|---|---|---|---|---|---|
| | | None | Present | OR | 95% CI | p-value | Adjusted OR | 95% CI | p-value |
| Stapler size | 28/29 mm | 102 (96.2%) | 4 (3.8%) | 0.49 | 0.09–2.74 | 0.683 | | | |
| | 25 mm | 104 (98.1%) | 2 (1.9%) | | | | | | |
| Anastomotic leakage | None | 204 (98.1%) | 4 (1.9%) | 51.00 | 5.68–458.25 | **0.004** | 39.38 | 2.49–623.36 | 0.009 |
| | Present | 2 (50.0%) | 2 (50.0%) | | | | | | |
| Age (years) | < 70 | 130 (97.0%) | 4 (3.0%) | 0.86 | 0.15–4.78 | > 0.999 | | | |
| | ≥ 70 | 76 (97.4%) | 2 (2.6%) | | | | | | |
| Sex | Male | 101 (97.1%) | 3 (2.9%) | 0.96 | 0.19–4.88 | > 0.999 | | | |
| | Female | 105 (97.2%) | 3 (2.8%) | | | | | | |
| Body mass index (kg/m$^2$) | < 25 | 128 (97.0%) | 4 (3.0%) | 0.82 | 0.15–4.59 | > 0.999 | | | |
| | ≥ 25 | 78 (97.5%) | 2 (2.5%) | | | | | | |
| Diabetes | None | 158 (96.9%) | 5 (3.1%) | 0.66 | 0.08–5.78 | > 0.999 | | | |
| | Present | 48 (98.0%) | 1 (2.0%) | | | | | | |
| Hypertension | None | 109 (95.6%) | 5 (4.4%) | 0.23 | 0.03–1.96 | 0.22 | | | |
| | Present | 97 (99.0%) | 1 (1.0%) | | | | | | |
| Heart disease | None | 187 (96.9%) | 6 (3.1%) | 0.97 | 0.95–0.99 | > 0.999 | | | |
| | Present | 19 (100%) | 0 | | | | | | |
| Pulmonary disease | None | 189 (97.4%) | 5 (2.6%) | 2.22 | 0.25–20.14 | 0.417 | | | |
| | Present | 17 (94.4%) | 1 (5.6%) | | | | | | |
| Liver disease | None | 200 (97.1%) | 6 (2.9%) | 0.97 | 0.95–0.99 | > 0.999 | | | |
| | Present | 6 (100%) | 0 | | | | | | |
| Cerebrovascular disease | None | 196 (98.0%) | 4 (2.0%) | 9.80 | 1.60–60.03 | 0.039 | 7.22 | 0.76–68.58 | 0.085 |
| | Present | 10 (83.3%) | 2 (16.7%) | | | | | | |
| ASA classification | 1, 2 | 166 (97.1%) | 5 (2.9%) | 0.83 | 0.09–7.30 | > 0.999 | | | |
| | 3, 4 | 40 (97.6%) | 1 (2.4%) | | | | | | |
| Smoking history | None | 156 (97.5%) | 4 (2.5%) | 1.56 | 0.28–8.77 | 0.637 | | | |
| | Present | 50 (96.2%) | 2 (3.8%) | | | | | | |
| Alcoholic history | None | 157 (97.5%) | 4 (2.5%) | 1.60 | 0.29–9.01 | 0.632 | | | |
| | Present | 49 (96.1%) | 2 (3.9%) | | | | | | |
| Emergency or elective | Elective | 174 (96.7%) | 6 (3.3%) | 0.97 | 0.94–0.99 | 0.594 | | | |
| | Emergency | 32 (100%) | 0 | | | | | | |
| Benign lesion or cancer | Benign | 68 (97.1%) | 2 (2.9%) | 0.99 | 0.18–5.52 | > 0.999 | | | |
| | Cancer | 138 (97.2%) | 4 (2.8%) | | | | | | |
| Anastomosis type | Ileocolic | 107 (100%) | 0 | | | | | | |
| | Colorectal | 99 (94.3%) | 6 (5.7%) | 1.06 | 1.01–1.11 | 0.014 | | | |

Abbreviation: ASA, American Society of Anesthesiologists; OR, Odd ratios; CI, confidence interval

In one patient using a 28-mm circular stapler, stricture was confirmed 95 days after surgery, and there was no further stricture after 2485 days with a single finger dilatation. In one patient who used a 25-mm circular stapler, stricture was confirmed 157 days after surgery, but there was no further stricture for 385 days after one finger dilatation.

Patients who underwent left hemicolectomy using a 28-mm circular stapler developed anastomotic stricture after 60 days, and a stent was inserted to resolve the stricture. After stent insertion, symptoms improved, and the patients were discharged; however, follow-up was lost thereafter.

## Discussion

Our study shows that when performing anastomosis using circular staplers in colorectal surgery, there was no difference in the incidence of anastomotic strictures between patients in whom the 25-mm circular staplers were used and patients in whom the 28/29-mm circular staplers were used. It is worth noting that colorectal anastomosis had a rate of higher stricture formation than ileocolic anastomosis, and anastomotic leakage was confirmed to be a statistically significant factor affecting the formation of anastomotic strictures.

The use of staplers for performing intestinal anastomosis during colorectal surgery has the advantage of being convenient and shortening the operation time compared to performing handsewn anastomosis [14]. However, as with handsewn anastomosis, complications associated with surgery are unavoidable. Anastomotic stricture is one such common anastomotic complication, reported in 3%–30% of cases [15, 16].

There is debate regarding the effect of stapler size on anastomotic strictures. According to a meta-analysis published in 2018 on gastrointestinal (GI) anastomotic strictures with respect to circular stapler size, after analyzing three lower GI studies, it was reported that the incidence of anastomotic stricture increased when 25-mm circular staplers were used compared to when 28/29-mm circular staplers were used (OR: 2.60, 95% CI: 0.82–8.29, p = 0.10) [6]. However, the data that was assessed in this meta-analysis mostly included patients with pouch-anal anastomosis for the treatment of familial adenomatous polyposis or inflammatory bowel disease. In addition, in 2020, Reif de Paula et al. reported that the smaller the circular stapler size, the more frequent the formation of strictures, with an OR of 3.482 [95% CI: 1.078–11.247, p = 0.037] [11]. However, the stapler sizes used as a part of that particular study were 28–29 mm and 31–33 mm.

A recent study showed that the size of circular staplers did not affect the rate of formation of strictures. In 2021, Nagoaka et al. compared 25-mm and 28/29-mm stapler use and confirmed that stapler size was not related to anastomotic stricture incidence. (OR: 1.7, 95% CI: 0.7–3.8, p = 0.21) [17]. Our study also showed that circular stapler size did not affect anastomotic stricture incidence (OR: 0.49, 95% CI: 0.09–2.74, p = 0.683); therefore, stapler size seems to have no effect on anastomotic stricture formation rates. In our study, the only factor affecting anastomotic stricture formation was anastomotic leakage (adjusted: OR 39.38, 95% CI: 2.49–623.36, p = 0.009), which is consistent with previously published literature [15, 17, 18]. Leakage-induced pelvic inflammation may be a major cause of anastomotic stenosis [8].

Relatively smaller circular staplers have the advantage of being easier to insert through the colon, rectum, or anus when performing ileocolic or colorectal anastomosis. In particular, in the case of narrow colonic lumens, wherein it is difficult to insert a 28/29-mm stapler, the excessive force required for stapler insertion may inflict unnecessary colonic injury. If the anal canal or rectum is narrow, forcefully inserting a large circular stapler may injure the rectum, levator ani muscles, or surrounding structures, such as the vagina [4, 19]. Although there was no incidence of iatrogenic injury caused by the use of 28/29-mm circular staplers in our study, there seems to be no need to use larger circular staplers because a small stapler seems to be adequate and does not increase the risk of harm.

A variety of methods have been reported for the resolution of anastomotic strictures, ranging from manual dilatation and endoscopic procedures to re-anastomosis [15, 20–22]. Strictures that are accessible up to the line of anastomosis via the anus can be resolved by manual finger dilatation [17]. In our study, two cases of anastomotic stricture that occurred following colorectal surgery improved with manual finger dilatation itself. Stent insertion was performed in one patient who developed anastomotic stricture following left hemicolectomy because life

expectancy was expected to be suboptimal, owing to underlying stage-IV peritoneal seeding with malignant ascites.

Although there are recently published papers on the safety of end-to-side ileocolonic anastomosis [23, 24], few studies have reported the development of anastomotic strictures limited to the ileocolic end-to-side anastomosis. In our study, when performing ileocolic end-to-side anastomosis using a circular stapler following ileocolectomy, there was no case of anastomotic stricture. However, since the number of patients who underwent ileocolic anastomosis aided by 25-mm circular staplers in our study was only 55, additional large-scale studies in the future are required to verify the safety of using a small stapler for performing end-to-side anastomosis.

Our study has certain limitations. Firstly, the possibility of selection bias was not eliminated, as it is a retrospective study conducted at a single center. Above all, the type of stapler used depends on the surgeon's decision, and therefore, the basis for the selection of a 25-mm or 28/29-mm was unknown. Moreover, in some cases, stricture occurred despite the absence of leakage, which reiterates the need for a randomized control study using more samples. Secondly, we could not confirm the rate of incidence of strictures in patients who did not undergo colonoscopy. Furthermore, even the patients who did undergo colonoscopy, did not undergo the procedure at regular intervals. Anastomotic stricture is usually reported to occur 5–12 months following anastomosis performed during surgery [16, 22, 25]. In particular, 37 of the 210 patients (17.6%) for whom no stricture was found had their first anastomosis confirmed through colonoscopy within 5 months. Therefore, it cannot be excluded that a stricture may develop much later in the patients who underwent colonoscopy only within a 5-month period after surgery. Finally, it is difficult to generalize the results of our study because there was no heterogeneity of race in our sample population, as it consisted of Asian patients alone, whose builds and physiques are unique.

## Conclusion

Circular stapler size does not influence anastomotic stricture formation in colorectal surgery. As this was a retrospective study conducted at a single institution, it is necessary to pursue this line of research via multicenter prospective studies in the future. Hopefully, that will help guide surgeons regarding appropriate staple size selection in colorectal surgeries.

## Supporting information

**S1 File.**
(DOCX)

## Author Contributions

**Conceptualization:** Kil-yong Lee, Jaeim Lee.

**Data curation:** Kil-yong Lee.

**Formal analysis:** Kil-yong Lee, Jaeim Lee, Youn Young Park, Hyung-Jin Kim, Seong Taek Oh.

**Investigation:** Kil-yong Lee.

**Methodology:** Kil-yong Lee, Jaeim Lee.

**Project administration:** Kil-yong Lee.

**Resources:** Kil-yong Lee.

**Supervision:** Kil-yong Lee, Jaeim Lee.

**Validation:** Kil-yong Lee, Jaeim Lee.

**Writing – original draft:** Kil-yong Lee.

**Writing – review & editing:** Kil-yong Lee, Jaeim Lee, Youn Young Park, Hyung-Jin Kim, Seong Taek Oh.

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
