## [Decision Letter · Decision Letter 0]

27 Feb 2023

PONE-D-23-02247The Effect of Circular Stapler Size on Anastomotic Stricture Formation in Colorectal Surgery: A Propensity Score Matched StudyPLOS ONE Dear Dr. Lee,

Thank you for submitting your manuscript to PLOS ONE. After careful consideration, we feel that it has merit but does not fully meet PLOS ONE’s publication criteria as it currently stands. Therefore, we invite you to submit a revised version of the manuscript that addresses the points raised during the review process.

We look forward to receiving your revised manuscript.

Kind regards,

Fabrizio D'Acapito, Ph.D,M.D.

Academic Editor

PLOS ONE

Journal Requirements:

2. In the ethics statement in the manuscript and in the online submission form, please provide additional information about the patient records/samples used in your retrospective study. Specifically, please ensure that you have discussed whether all data/samples were fully anonymized before you accessed them and/or whether the IRB or ethics committee waived the requirement for informed consent. If patients provided informed written consent to have data/samples from their medical records used in research, please include this information.

3. Please provide further details about your study design. Specifically, please clarify whether patients were recruited prospectively or whether this study presents retrospective analyses of patient medical records.

4. Please provide additional details regarding participant consent. In the ethics statement in the Methods and online submission information, please ensure that you have specified what type you obtained (for instance, written or verbal, and if verbal, how it was documented and witnessed). If your study included minors, state whether you obtained consent from parents or guardians. If the need for consent was waived by the ethics committee, please include this information.

"No. The funders had no role in study design, data collection and analysis, decision to publish, or preparation of the manuscript."

6. We note that you have indicated that data from this study are available upon request. PLOS only allows data to be available upon request if there are legal or ethical restrictions on sharing data publicly. For more information on unacceptable data access restrictions, please see http://journals.plos.org/plosone/s/data-availability#loc-unacceptable-data-access-restrictions. 

**Additional Editor Comments:**

I believe that the reviewers have focused on the weaknesses of the paper, while also providing insights to correct them.

I think some of the reviewers' considerations play a crucial role in meaningfully improving the paper:

“….Since the risk profile for anastomotic stenosis and also anastomotic leaks differs significantly from ileocolic anastomoses and colorectal anastomoses, I consider the selection of the patient collective to be unsuitable. As confirmed in your data, only an evaluation of the colorectal anastomoses makes sense.

… I suggest to include only those patients who underwent left colectomy or anterior resection, ….In addition, the impact of a derivative stoma should be taken in consideration during the analysis of the results.

…. please could the Authors specify why the circular stapler sized > 29-mm (widely used in rectal cancer surgery) have been excluded.

… Who and why chooses a 25-mm or 28/29-mm stapler?”

An improvement in English would be desirable.

Reviewers' comments:

Reviewer's Responses to Questions

**Comments to the Author**

1. Is the manuscript technically sound, and do the data support the conclusions?

Reviewer #1: Partly

Reviewer #2: Partly

Reviewer #3: No

2. Has the statistical analysis been performed appropriately and rigorously? 

Reviewer #1: Yes

Reviewer #2: Yes

Reviewer #3: Yes

3. Have the authors made all data underlying the findings in their manuscript fully available?

Reviewer #1: Yes

Reviewer #2: Yes

Reviewer #3: Yes

4. Is the manuscript presented in an intelligible fashion and written in standard English?

Reviewer #1: Yes

Reviewer #2: Yes

Reviewer #3: Yes

5. Review Comments to the Author

Reviewer #1: The authors report the effect of cicular stapler size on anastomotic stricture in colorectal surgery.

This is a relevant question in colorectal surgery and therefore an interesting topic. Overall, the conduct of the study is correct and the manuscript is in order. An improvement in English would be desirable.

However, I have one major concern:

Since the risk profile for anastomotic stenosis and also anastomotic leaks differs significantly from ileocolic anastomoses and colorectal anastomoses, I consider the selection of the patient collective to be unsuitable. As confirmed in your data, only an evaluation of the colorectal anastomoses makes sense.

Reviewer #2: Thanks to the Authors for this interesting paper.

I would like to make the following considerations and questions.

METHODS

Patients

The Authors assert that the purpose of the study is to determine whether the frequency of anastomotic stricture depends on the stapler size: please could the Authors specify why the circular stapler sized > 29-mm (widely used in rectal cancer surgery) have been excluded.

Procedure

Do the Authors usually or always perform end-to-side anastomosis by mean a circular stapler following ileocecal resection or right colectomy? Is a linear stapler used for section or closure of any bowel stump?

Who and why chooses a 25-mm or 28/29-mm stapler?

Statistical analyses

Please could the Authors specify the methodology for after matching analysis, for example McNemar’s test, paired t test or Wilcoxon signed rank test.

RESULTS

Baseline characteristics

In table 1 the clinical characteristics of patients are similar but 25-mm staplers are significantly more used in benign lesions (28.3% vs 8.1% of 28/29-mm stapler group) and emergency surgery (15.1% vs 4.8% of 28/29-mm stapler group) whilst 25-mm staplers are significantly less used in colorectal anastomosis (1 out of 23 – 4.3% – left colectomy patients and 47 out of 654 – 7.2% – anterior resection patients): how do the Authors motivate these so impressive differences?

Outcomes

Anastomotic stenosis occurred in 6 cases, 4 in the 28/29-mm group (in 2 cases after anastomotic leak) and 2 in the 25-mm group (without any leak) and in table 3 only anastomotic leak resulted as independent risk factor for stricture at multivariate analysis: despite the small number of cases, is there any reason for stricture in patients without leak, especially in the 2 patients in the 25-mm group?

Anastomotic stricture management

Please could the Authors specify which stapler was used in the 3 patients that underwent operative treatment for symptomatic stricture.

Was the decision of stent insertion in the patient with stenosis after left colectomy dictated by the specific oncological stage of the disease or is stent insertion part of the armamentarium of strictures’ treatment?

How many finger dilatations were performed in order to obtain a permanently wide stricture site? Was pneumatic dilatation considered?

Please could the Authors provide details concerning follow up duration and long-term efficacy of treatment.

DISCUSSION

The Authors report that the 2011 Cochrane Database Systematic Review (reference 1) showed no differences in incidence of stricture following stapled anastomosis, compared with hand-sewn anastomosis, and correctly underline the small number of patients (65) included in the two studies evaluating the outcome “anastomotic stricture” (Izbicki JR et al. Der Chirurg 1998;69:725-34, published data only; Ikeuchi H et al. Digestive Surgery 2000;17:493-496, published and unpublished data); however, all the 26 “stapled” patients included in the two above mentioned studies underwent a side-to-side ileocolic anastomosis by mean of a linear and not circular stapler, therefore these data don't seem to have a relationship with the aim of the study.

Reviewer #3: This is a manuscript that aims at evaluating the effect of circular stapler size on anastomotic stricture formation in colorectal surgery. The authors found no significant differences between the two groups. However, the interpretation of the results is biased by the small number of patients included in the two groups (the sample size calculation is not mentioned) and by the inclusion of both right and left colectomies/anterior resections. I suggest to include only those patients who underwent left colectomy or anterior resection, since the incidence of anastomotic stricture is negligible after right colectomy. In addition, the impact of a derivative stoma should be taken in consideration during the analysis of the results.

Lastly, the timing of anastomotic check is not clearly reported, as also the clinical relevance of the strictures (did the patients with stricture report any symptom? From the paragraph assessing the stenosis management, it seems the the clinical impact was minimal.

Minor:

> please number the pages

> some English editing is recommended

6. PLOS authors have the option to publish the peer review history of their article (what does this mean?). If published, this will include your full peer review and any attached files.

Reviewer #1: No

Reviewer #2: No

Reviewer #3: **Yes: **Marco Ettore Allaix

---

## [Author Response · Author response to Decision Letter 0]

19 Apr 2023

Reviewer #1: The authors report the effect of cicular stapler size on anastomotic stricture in colorectal surgery.

This is a relevant question in colorectal surgery and therefore an interesting topic. Overall, the conduct of the study is correct and the manuscript is in order. An improvement in English would be desirable.

However, I have one major concern:

Since the risk profile for anastomotic stenosis and also anastomotic leaks differs significantly from ileocolic anastomoses and colorectal anastomoses, I consider the selection of the patient collective to be unsuitable. As confirmed in your data, only an evaluation of the colorectal anastomoses makes sense.

Answer>

We thank the reviewer for their comments. 

As per the reviewer’s recommendation, we performed a subgroup analysis for 105 patients with colorectal anastomosis. Even in this subgroup analysis, there was no difference in anastomotic stricture between the two groups.

We described the relevant procedure in the revised manuscript (line 170) and included the data in supplemental tables (S1-S3).

We revised the manuscript to improve the language.

 

Reviewer #2: Thanks to the Authors for this interesting paper.

I would like to make the following considerations and questions.

METHODS

Patients

The Authors assert that the purpose of the study is to determine whether the frequency of anastomotic stricture depends on the stapler size: please could the Authors specify why the circular stapler sized > 29-mm (widely used in rectal cancer surgery) have been excluded.

Answer>

We appreciate your considerate comments. In our study data, only 12 (1.3%) out of a total of 894 patients used 29 mm or more. Additionally, in a study with a similar subject to ours1,2, a single group of 28/29-mm was used. Therefore, the group of “29 mm or more” was excluded. The number of patients corresponding to each stapler size in the excluded patient group has been added to Fig 1.

Refereces>

1. T. Nagaoka et al. Dis Colon Rectum. 2021 Aug 1;64(8):937-945.

2. T. Reif de Paula et al. Tech Coloproctol. 2020 Apr;24(4):283-290.

Procedure

Do the Authors usually or always perform end-to-side anastomosis by mean a circular stapler following ileocecal resection or right colectomy? Is a linear stapler used for section or closure of any bowel stump?

Who and why chooses a 25-mm or 28/29-mm stapler?

Answer>

At our institution, end-to-side anastomosis is always performed when performing ileocecal resection or right hemicolectomy, and a linear stapler is used for stump closure.

25-mm or 28/29-mm staplers are used entirely according to the surgeon's decision; therefore, the exact reason for this selection is unknown. We have added the relevant explanations to the procedure (line 85) and limitation sections (line 253) of the revised manuscript.

Statistical analyses

Please could the Authors specify the methodology for after matching analysis, for example McNemar’s test, paired t test or Wilcoxon signed rank test.

Answer>

After propensity score matching, the clinical characteristics were analyzed using the Mann–Whitney U test or Student’s t-test for continuous variables and the chi-squared test, Fisher’s exact test, or linear-by-linear association for categorical variables. A series of analyses related to propensity score matching was conducted using Web-based Analysis with R (prepared by web-r.org) based on the R package MatchIt. We have added this description to the methods section (line 111).

RESULTS

Baseline characteristics

In table 1 the clinical characteristics of patients are similar but 25-mm staplers are significantly more used in benign lesions (28.3% vs 8.1% of 28/29-mm stapler group) and emergency surgery (15.1% vs 4.8% of 28/29-mm stapler group) whilst 25-mm staplers are significantly less used in colorectal anastomosis (1 out of 23 – 4.3% – left colectomy patients and 47 out of 654 – 7.2% – anterior resection patients): how do the Authors motivate these so impressive differences?

Answer>

The reason for the occurrence of this discrepancy is unknown, as the size of the round stapler used was entirely at the discretion of the surgeon. However, to compensate for this difference, we performed a propensity score matching.

Outcomes

Anastomotic stenosis occurred in 6 cases, 4 in the 28/29-mm group (in 2 cases after anastomotic leak) and 2 in the 25-mm group (without any leak) and in table 3 only anastomotic leak resulted as independent risk factor for stricture at multivariate analysis: despite the small number of cases, is there any reason for stricture in patients without leak, especially in the 2 patients in the 25-mm group?

Answer>

Due to the small number of patients included in our study, determining the reason for stricture occurrence in the 25-mm group without the risk factors from our study was not possible. However, considering that stricture occurred in two patients in the 28/29-mm group without leakage, we believe that there was a reason other than the stapler size. Additional research is needed to elucidate this point; therefore, we have included it in the limitations section (line 255).

Anastomotic stricture management

Please could the Authors specify which stapler was used in the 3 patients that underwent operative treatment for symptomatic stricture.

Was the decision of stent insertion in the patient with stenosis after left colectomy dictated by the specific oncological stage of the disease or is stent insertion part of the armamentarium of strictures’ treatment?

How many finger dilatations were performed in order to obtain a permanently wide stricture site? Was pneumatic dilatation considered?

Please could the Authors provide details concerning follow up duration and long-term efficacy of treatment.

Answer>

In one patient using a 28-mm circular stapler, stricture was confirmed 95 days after surgery, and there was no further stricture after 2485 days with a single finger dilatation.

In one patient who used a 25-mm circular stapler, stricture was confirmed 157 days after surgery, but there was no further stricture for 385 days after one finger dilatation.

Patients who underwent left hemicolectomy using a 28-mm circular stapler developed anastomotic stricture after 60 days, and a stent was inserted to resolve the stricture. After stent insertion, symptoms improved, and the patients were discharged; however, follow-up was lost thereafter. When the medical records were checked, pneumatic dilatation was not considered for this patient.

We have added these details to the revised manuscript (line 186).

DISCUSSION

The Authors report that the 2011 Cochrane Database Systematic Review (reference 1) showed no differences in incidence of stricture following stapled anastomosis, compared with hand-sewn anastomosis, and correctly underline the small number of patients (65) included in the two studies evaluating the outcome “anastomotic stricture” (Izbicki JR et al. Der Chirurg 1998;69:725-34, published data only; Ikeuchi H et al. Digestive Surgery 2000;17:493-496, published and unpublished data); however, all the 26 “stapled” patients included in the two above mentioned studies underwent a side-to-side ileocolic anastomosis by mean of a linear and not circular stapler, therefore these data don't seem to have a relationship with the aim of the study.

Answer>

As per the reviewer’s comment, we have deleted the relevant section.

 

Reviewer #3: This is a manuscript that aims at evaluating the effect of circular stapler size on anastomotic stricture formation in colorectal surgery. The authors found no significant differences between the two groups. However, the interpretation of the results is biased by the small number of patients included in the two groups (the sample size calculation is not mentioned) and by the inclusion of both right and left colectomies/anterior resections. I suggest to include only those patients who underwent left colectomy or anterior resection, since the incidence of anastomotic stricture is negligible after right colectomy. In addition, the impact of a derivative stoma should be taken in consideration during the analysis of the results.

Lastly, the timing of anastomotic check is not clearly reported, as also the clinical relevance of the strictures (did the patients with stricture report any symptom? From the paragraph assessing the stenosis management, it seems the the clinical impact was minimal.

Answer>

We thank the reviewer for their considerate comments.

Following the reviewer’s recommendation, we performed a subgroup analysis for 105 patients with colorectal anastomosis and evaluated the effect of diversion (stoma) on anastomotic stricture (added: line 170; Table S1, S3).

Finally, we confirmed anastomotic stricture with colonoscopy for all patients (added: line 89).

---

## [Decision Letter · Decision Letter 1]

8 Jun 2023

The Effect of Circular Stapler Size on Anastomotic Stricture Formation in Colorectal Surgery: A Propensity Score Matched Study

PONE-D-23-02247R1

Dear Dr. Lee,

We’re pleased to inform you that your manuscript has been judged scientifically suitable for publication and will be formally accepted for publication once it meets all outstanding technical requirements.

Kind regards,

Fabrizio D'Acapito, Ph.D,M.D.

Academic Editor

PLOS ONE

Additional Editor Comments (optional):

I congratulate the authors for their review work. The topic has important clinical relevance for those who deal with colo-rectal resective surgery on a daily basis.

The input that the paper provides may be of assistance to the surgeon and a boost to launch further research in this field.

Reviewers' comments:

Reviewer's Responses to Questions

**Comments to the Author**

1. If the authors have adequately addressed your comments raised in a previous round of review and you feel that this manuscript is now acceptable for publication, you may indicate that here to bypass the “Comments to the Author” section, enter your conflict of interest statement in the “Confidential to Editor” section, and submit your "Accept" recommendation.

Reviewer #3: All comments have been addressed

2. Is the manuscript technically sound, and do the data support the conclusions?

Reviewer #3: Yes

3. Has the statistical analysis been performed appropriately and rigorously? 

Reviewer #3: Yes

4. Have the authors made all data underlying the findings in their manuscript fully available?

Reviewer #3: Yes

5. Is the manuscript presented in an intelligible fashion and written in standard English?

Reviewer #3: Yes

6. Review Comments to the Author

Reviewer #3: The authors have replied to the questions point-by-point. All comments have been adequately assessed.

7. PLOS authors have the option to publish the peer review history of their article (what does this mean?). If published, this will include your full peer review and any attached files.

Reviewer #3: No

---

## [Editor Report · Acceptance letter]

15 Jun 2023

PONE-D-23-02247R1 

The Effect of Circular Stapler Size on Anastomotic Stricture Formation in Colorectal Surgery: A Propensity Score Matched Study 

Dear Dr. Lee:

I'm pleased to inform you that your manuscript has been deemed suitable for publication in PLOS ONE. Congratulations! Your manuscript is now with our production department. 

Kind regards, 

on behalf of

Dr. Fabrizio D'Acapito 

Academic Editor

PLOS ONE